# Sugar-based bactericides targeting phosphatidylethanolamine-enriched membranes

Catarina Dias[1,2], João P. Pais[1,2,3], Rafael Nunes [1,2,3], Maria-Teresa Blázquez-Sánchez[1,2], Joaquim T. Marquês[1,2], Andreia F. Almeida [1], Patrícia Serra[1], Nuno M. Xavier[1,2], Diogo Vila-Viçosa[1,3], Miguel Machuqueiro [1,3], Ana S. Viana[1,2], Alice Martins[1,2], Maria S. Santos[1,2], Ana Pelerito[4], Ricardo Dias[3], Rogério Tenreiro[3], Maria C. Oliveira[2], Marialessandra Contino[5], Nicola A. Colabufo[5,6], Rodrigo F.M. de Almeida[1,2] & Amélia P. Rauter [1,2]

Anthrax is an infectious disease caused by *Bacillus anthracis*, a bioterrorism agent that develops resistance to clinically used antibiotics. Therefore, alternative mechanisms of action remain a challenge. Herein, we disclose deoxy glycosides responsible for specific carbohydrate-phospholipid interactions, causing phosphatidylethanolamine lamellar-to-inverted hexagonal phase transition and acting over *B. anthracis* and *Bacillus cereus* as potent and selective bactericides. Biological studies of the synthesized compound series differing in the anomeric atom, glycone configuration and deoxygenation pattern show that the latter is indeed a key modulator of efficacy and selectivity. Biomolecular simulations show no tendency to pore formation, whereas differential metabolomics and genomics rule out proteins as targets. Complete bacteria cell death in 10 min and cellular envelope disruption corroborate an effect over lipid polymorphism. Biophysical approaches show monolayer and bilayer reorganization with fast and high permeabilizing activity toward phosphatidylethanolamine membranes. Absence of bacterial resistance further supports this mechanism, triggering innovation on membrane-targeting antimicrobials.

[1] Centro de Química e Bioquímica, Faculdade de Ciências, Universidade de Lisboa, Ed. C8, Campo Grande, 1749-016 Lisboa, Portugal. [2] Centro de Química Estrutural, Faculdade de Ciências, Universidade de Lisboa, 1749-016 Lisboa, Portugal. [3] Biosystems and Integrative Sciences Institute, Faculdade de Ciências, Universidade de Lisboa, Campo Grande, 1749-016 Lisboa, Portugal. [4] Instituto Nacional de Saúde Doutor Ricardo Jorge, Av. Padre Cruz, 1649-016 Lisboa, Portugal. [5] Dipartimento di Farmacia-Scienze del Farmaco, Università degli Studi di Bari, Via Edoardo Orabona, 4, 70125 Bari, Italy. [6] Biofordrug/Università degli Studi di Bari, Via Edoardo Orabona, 4, 70125 Bari, Italy. These authors contributed equally: Catarina Dias, João P. Pais, Rafael Nunes. Correspondence and requests for materials should be addressed to A.P.R. (email: aprauter@fc.ul.pt)

Anthrax is an infectious disease affecting livestock and humans who handle infected farm animals. In addition, this species is a serious bioterrorism threat[1,2]. All types of anthrax have the potential, if untreated, to spread throughout the body causing severe illness and death. Regrettably, therapeutic options for anthrax are insufficient, relying on long-term, intravenous, combined antibiotic treatment with quinolones and tetracyclines[2]. The disease is caused by *Bacillus anthracis*, a Gram-positive bacterium that develops resistance to the antibiotics of choice, namely ciprofloxacin, and examples of new antibacterial molecules against *B. anthracis* are scarce[3–6]. The microbe genetically closely related to *B. anthracis* is *Bacillus cereus*[7], commonly used for research on antimicrobials against *B. anthracis*, as it can be manipulated in laboratories of safety level 2 (BSL2). This pathogen is responsible for food poisoning, and for skin and eye infections, although recently it has drawn more attention as it has been reported to form biofilms in medical materials[8]. This feature is particularly worrisome since biofilm forming bacteria have increased capacity to survive hospital environments and implanted medical devices, thriving chronic wounds, persistent infections and creating a barrier to the immune system and to antibiotics[8].

Our preliminary data on the antimicrobial activity of the sugar-based surfactants dodecyl 2,6-dideoxy *arabino*-hexopyranoside and its 2-deoxy analog, detected by the paper disc diffusion method over *B. cereus*, have shown that only the 2,6-dideoxy glycoside acts as a potent and selective antimicrobial over *Bacillus* spp.[6]. The observed selectivity does not fit the typical behavior of glucoside surfactants[9], also reported as permeability enhancers at concentrations over 190 μM on human cell lines, allowing significant cell recovery due to negligible membrane disruption[10].

In this study, we explore the family of deoxy glycosides to unravel their antibacterial mechanism of action, through generation by synthesis of a small library of deoxy glycosides and analogs, structure–activity relationship and mechanistic studies. Structural features, namely atom linking the dodecyl chain to the sugar (C–O vs. C–S vs. C–C bond), deoxygenation pattern (2-deoxy, 4-deoxy, 6-deoxy, 2,6-dideoxy, 4,6-dideoxy), sugar configuration (*arabino*, *threo*, *lyxo*, and *manno*), anomeric configuration, and hexopyranoside vs. pentopyranoside structure, are investigated. The antibacterial profiling and the bioactivity on *B. cereus* spore germination are evaluated. Target assessment and genetic studies are carried out, as well as simulation studies aiming to rationalize the behavior of deoxy glycosides. Considering that *B. cereus* bacterial membrane is rich in phosphatidylethanolamine (PE), unlike the outer leaflet of mammalian cells, which is rich in phosphatidylcholine (PC), biophysical studies on PE-rich vs. PC-rich membranes are also conducted. This interdisciplinary approach enabled us to disclose the mechanism of action of these neutral sugar-based bactericides targeting membrane PE, and thereby promoting bacterial membrane disruption through phospholipid lamellar-to-inverted hexagonal phase transition.

## Results

**Synthesis.** A small library of dodecyl deoxy glycosides differing by the atom (O, S, C) linking the alkyl chain to the glycone, deoxygenation pattern, anomeric, and glycone configuration (*arabino*, *threo*, *lyxo*, and *manno*), and bearing a hexoside/pentoside structure is generated by synthesis. Dodecyl 2-deoxy *O*- and *S*-glycosides from both D and L series, namely compounds **2**, **4**, **7**, **8**, **17–20** (Fig. 1), are accessed by the proven method based on reaction of glycals with dodecan-1-ol or dodecane-1-thiol, catalyzed by triphenylphosphane hydrobromide[11], followed by deprotection (Supplementary Fig 1), also applied to synthesize compounds **1**, **5**, **9**, and **13**[6]. Noteworthy, pentopyranoside **8** adopted, in CDCl₃, the unusual ⁴C₁ conformation, as suggested by nuclear magnetic resonance (NMR) and rationalized by density functional theory (DFT) calculations (Supplementary Discussion, Supplementary Figs 2 and 3, and Supplementary Table 1).

In order to disclose the biological impact of 2-deoxygenation, 6-deoxy-*manno* enantiomers **6** and **16** (Fig. 1) are prepared starting from L-rhamnose and D-mannose, respectively, in good overall yield (Supplementary Fig. 4). Also 4,6-dideoxy-*xylo*-hexopyranosides **11** and **12**, and their precursor 4-deoxy-*xylo*-hexopyranoside **15**, are synthesized starting from a 4,6-*O*-benzylidene-protected glucopyranoside (Supplementary Fig. 5) in good overall yield.

Dodecyl 2,6-dideoxy *C*-glycoside **3** and its enantiomer **10**, along with the 2-deoxy *C*-hexopyranoside **14** and the *C*-pentopyranoside analog **21**, are accessed by the approach for which the key step is the metathesis reaction of allyl α-*C*-glycosides. These substrates were prepared via anomeric allylation[12], herein employed on deoxy sugar precursors, to

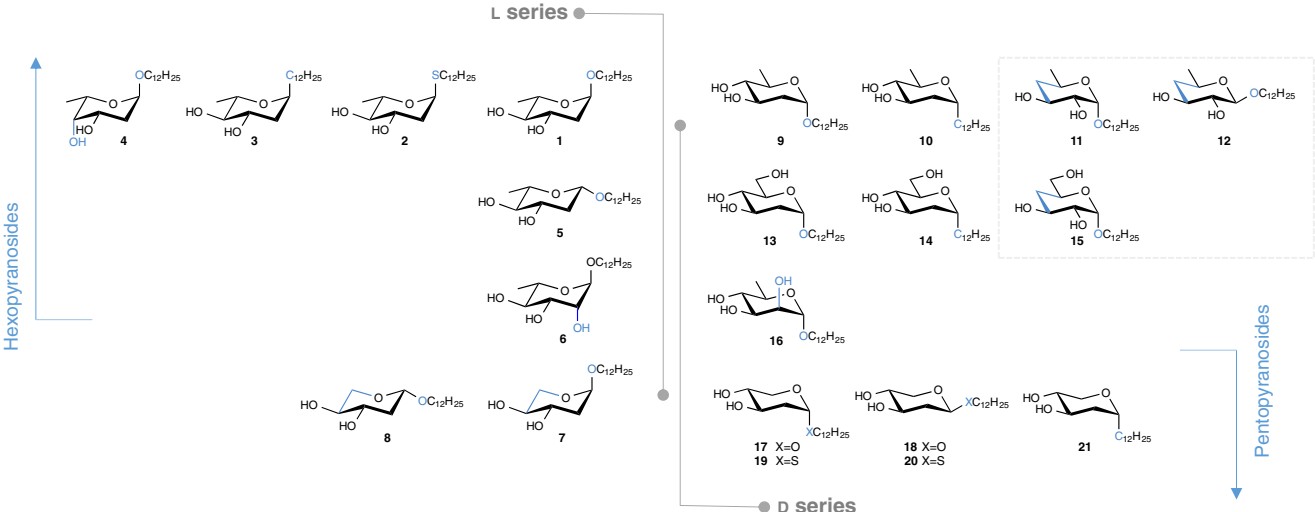

**Fig. 1** Lead series generated by chemical synthesis. Glycones differ in the deoxygenation pattern (2-deoxy, 4-deoxy, 6-deoxy, 2,6-dideoxy and 4,6-dideoxy), D- and L-configuration, anomeric configuration, and atom linkage to the dodecyl chain (details on synthesis are given in Supplementary Methods)

afford allyl α-*C*-glycosides in good yields. Their reaction with undec-1-ene in the presence of the second-generation Grubbs–Hoveyda catalyst (5 mol%) (Supplementary Fig 6), followed by double bond hydrogenation and Zemplén deacetylation gave the target molecules in high yield over the three steps. This efficient methodology opens the way to a diversity of *C*-hexopyranosides, which synthesis involves often drastic reaction conditions.

**Susceptibility against *B. anthracis* and cytotoxicity.** Reports on the mesomorphic behavior of alkyl 2-deoxy-α-D-*arabino*-hexopyranosides and 6-deoxy-β-D-glucopyranosides[13,14] encouraged us to unravel the impact of deoxy glycosides at the biological level. Their bactericide activity over *B. cereus*, *B. anthracis* (Sterne, pathogenic, and ovine), and *Enterococcus faecalis* is evaluated (Table 1). By comparing the bioactivity of hexopyranoside anomers of D or L series (pairs **1**, **5** and **11**, **12**), it becomes clear that α-configuration is crucial, corroborating our findings[6]. The deoxygenation patterns 6-deoxy, 2,6-dideoxy, and 4,6-dideoxy conduct to a high antimicrobial activity over all the bacteria tested, with the most promising result obtained for the 4,6-dideoxygenated glycoside **11**, presenting a minimum inhibitory concentration (MIC) fourfold lower than its 4-deoxy analog **15**. This tendency is kept for the 2-deoxy/2,6-dideoxy pair **14/10**, whereas compound **13** is not active over *B. cereus*, as expected, with a MIC value of 386 μM, also shown for *B. anthracis* Sterne. Interestingly, **13** is active over *B. anthracis* strains ovine and pathogenic. These out-of-trend results, which are not observed in all other 2- and 4-deoxy compounds studied, were not expected due to the genetic similarity of *B. anthracis* and *B. cereus*[7]. In general, C–C linkage benefits D-series bioactivity, whereas the relative configuration at C-3 and C-4 does not seem to have any impact. 2-Hydroxyglycosides **6** and **16** maintain the activity over *B. anthracis* Sterne and pathogenic strains, thus inferring that in hexopyranosides C-6 deoxygenation is crucial for activity over *B. cereus* and *B. anthracis* Sterne. MIC values over *E. faecalis* are mostly above 48 μM.

Compound cytotoxicity is assessed by evaluating the effect on Caco-2 cell viability by the MTT [3-(4,5-dimethylthiazol-2-yl)-2,5-diphenyltetrazolium bromide] assay (Table 1, Supplementary Fig 7). *C*-glycosides **3**, **10**, and **21**, and α-pentopyranosides **7** and **17** are the less toxic tested compounds, whereas 4,6-dideoxy glycoside **11** shows the highest selectivity for all tested bacteria vs. eukaryotic cells.

**Bactericidal activity and antimicrobial resistance.** Time-kill studies of *B. cereus* ATCC 14579 are performed to assess glycoside **1** bacteriostatic or bactericidal activity, in parallel to the generation of its analog library of compounds. The broth microdilution method according to Clinical and Laboratory Standards Institute (CLSI) guidelines[15] is chosen for the simultaneous data assessment by varying multiple experimental conditions, but the MIC values obtained are twofold higher than those above disclosed for the agar dilution method. The time-kill assay is accomplished with a temporal resolution of 10 min, the minimal time required for sampling, at the concentrations of 4–32 μg mL$^{-1}$ (Supplementary Fig 8, Supplementary Table 2). Although the resolution is far superior to the traditional method collecting results hourly, at $10^6$ starting cell concentration the kinetics of bacterial cell death cannot be detected at 16 μg mL$^{-1}$ compound concentration or higher, due to complete cell death, observed at the first temporal point. Still, a 1000-fold reduction ($T_{99.9}$) is reached within 60 min for initial bacterial load of $10^7$ and $10^8$ CFU mL$^{-1}$. No significant bacterial cell death is observed at 4 and 8 μg mL$^{-1}$ in comparison with controls. In all performed assays, MIC equal to minimum bactericidal concentration (MBC) values indicates bactericidal activity (Supplementary Table 3). The extremely fast bacterial death is also pointing towards a biophysical action at the cellular envelope.

**Table 1 Antibacterial activity[a] expressed in MICs and cytotoxicity over Caco-2 cells expressed in IC$_{50}$ (MTT assay)**

| Compound Nr | MIC (μM) | | | Cytotoxicity IC$_{50}$ (μM) | | | |
| | *B. anthracis* | | | *E. faecalis* | *B. cereus* | 24 h | 48 h |
| | Sterne | Pathogenic | Ovine | | | | |
| **1** | 25 | 25 | 25 | 50 | 25 | 100 | > 50 |
| **2** | 24 | 48 | 48 | > 48 | 48 | > 50 | > 50 |
| **3** | 27 | 54 | 54 | > 54 | 27 | >100 | 100 |
| **4** | 25 | 25 | 25 | > 50 | 25 | 100 | 50 |
| **5** | 202 | 101 | 101 | 101 | 202 | n.d. | n.d. |
| **6** | 24 | 24 | 48 | 48 | n.d. | n.d. | n.d. |
| **7** | 26 | 52 | 52 | 52 | n.d | >100 | 100 |
| **8** | 26 | 26 | 52 | 52 | n.d. | n.d. | n.d. |
| **9** | 50 | 50 | 50 | 50 | 25 | 100 | 50 |
| **10** | 27 | 27 | 27 | > 54 | 27 | >100 | >100 |
| **11** | 12.6 | 12.6 | 12.6 | 12.6 | 12.6 | n.d. | 50 |
| **12** | > 405 | > 405 | > 405 | > 405 | > 405 | n.d. | 50 |
| **13** | 386 | 12.0 | 12.0 | 12.0 | 386 | 100 | > 50 |
| **14** | 104 | 51 | 51 | 51 | 51 | n.d. | n.d. |
| **15** | 51 | 51 | 51 | 51 | 51 | n.d. | n.d. |
| **16** | 24 | 24 | > 48 | > 48 | 48 | n.d. | n.d. |
| **17** | 26 | 52 | 52 | > 52 | 52 | >100 | >100 |
| **18** | 26 | 26 | 52 | 52 | 52 | >100 | > 50 |
| **19** | 50 | 25 | 50 | 50 | 25 | n.d. | n.d. |
| **20** | 25 | 50 | 50 | > 50 | 50 | n.d. | n.d. |
| **21** | 28 | 28 | 28 | 56 | 28 | >100 | 100 |
| Ref.[b] | 25 | 25 | 25 | 19 | 19 | n.d. | n.d. |

[a]Müller–Hinton agar dilution method following CLSI guidelines[15]. [b]Chloramphenicol

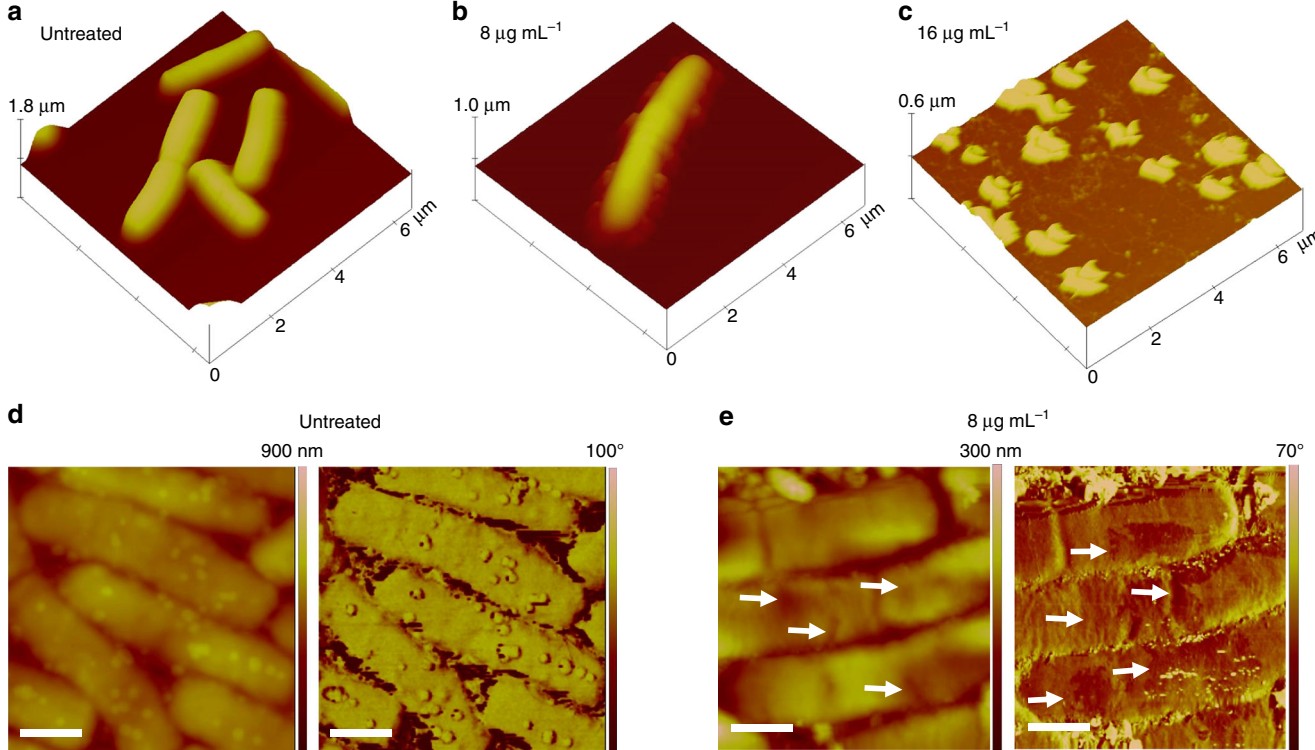

**Fig. 2** Tapping™ mode AFM imaging of *B. cereus* deposited on mica surface. **a**, **d** Depictions of *B. cereus* before the exposure of compound **1**, whereas **b**, **c**, and **e** show the bacteria in the presence of compound **1**. **a–c** 3D representations of topographic images, whereas **d** and **e** (scale bars = 750 nm) are detailed topographic (left panels) and phase-contrast (right panels) images of untreated cells and upon interaction with glycoside **1** (8 µg mL⁻¹), respectively. The arrows indicate lesions in the cellular envelope

The effect of compound **1** on *B. cereus* cells is also monitored by ex-situ atomic force microscopy (AFM) (Fig. 2). Three-dimensional topographic images clearly show the typical rod-shape expected for *B. cereus* and the complete loss of cellular integrity at MIC value. At sub-inhibitory concentrations, cellular damage by the presence of cracks and discontinuities at the cellular envelope is detected, in line with the bactericidal action of glycoside **1** by direct interaction with the cell envelope of *B. cereus*.

Resistance development, through natural mutagenesis, is assessed by submitting *B. cereus* to serial passage on gradient concentration of compound **1**. No alteration of MIC and MBC values is observed after 15 daily serial passages. Moreover, compound **1** antibacterial activity remains the same over *B. cereus* strains resistant to the main classes of known antibiotics (Supplementary Table 4), demonstrating that its mechanism of action is not related to that of those drugs and the mechanisms of resistance do not compromise compound activity. These findings show that this type of structures does not face the earlier emergence of bacterial resistance.

**Metabolomics, genetics, and ultrastructural studies.** Differential metabolomics analysis is carried out for detection of eventual interaction of compound **1** with cellular subsystems. By using Phenotype MicroArray-based approach[16,17], the differential analysis of *B. cereus* ATCC 14579 metabolism shows selective inhibition to several carbon sources. The associated metabolic pathways are identified according to the KEGG (Kyoto Encyclopedia of Genes and Genomes) database and hierarchical mapped (Supplementary Fig 9). Several clusters are identified, yielding as main targets the environmental information processing systems, namely ABC transport systems and phosphotransferases (PTS). Given the surfactant properties of these glycosides, differentiation between specific vs. non-specific mode of action can only be done by exclusion, since the surfactant effect is present. Hence, a genetic dissection for protein target assessment is carried out. A mutant library is created by random transposition, using *B. cereus* ATCC 14579 as isogenic system and its susceptibility to compound **1** is evaluated. MIC values remain unchanged when compared with the parental strain, indicating the absence of any resistance to the test compound, although a high number of mutants is obtained. In parallel, a library of specific knockout mutants for the several ABC and PTS systems previously identified as putative targets are challenged with compound **1** (Supplementary Table 5). Again, no difference between MIC values is observed in comparison with parental strain. These results suggest the absence of proteins as specific targets.

Aiming to understand the role of cellular envelope ultra-structure, protoplasts of *B. cereus*, *Staphylococcus aureus*, and spheroplasts of *Escherichia coli* K12 are submitted to the action of compound **1** (Fig. 3). The external cell wall of Gram-positive bacteria is not relevant since MIC values for bacteria and protoplasts are the same. *E. coli* spheroplast and *B. cereus* protoplast, containing 85% and 43% PE, respectively[18], have high susceptibility to **1** (Fig. 3, Supplementary Table 6), whereas no activity is found over *S. aureus* (PE ca. 0%), indicating that cytoplasmic membrane composition is crucial for activity. This phospholipid has also been detected in the composition of wild-type *B. anthracis* Sterne strain but it was not quantified[19]. It is assumed to be similar to that of *B. cereus*[18], to which it is closely related genetically[7].

*E. faecalis*, showing minor susceptibility, contains other zwitterionic lipids and unusual uncharged lipids in the cytoplasmic membrane, with biophysical properties different from those

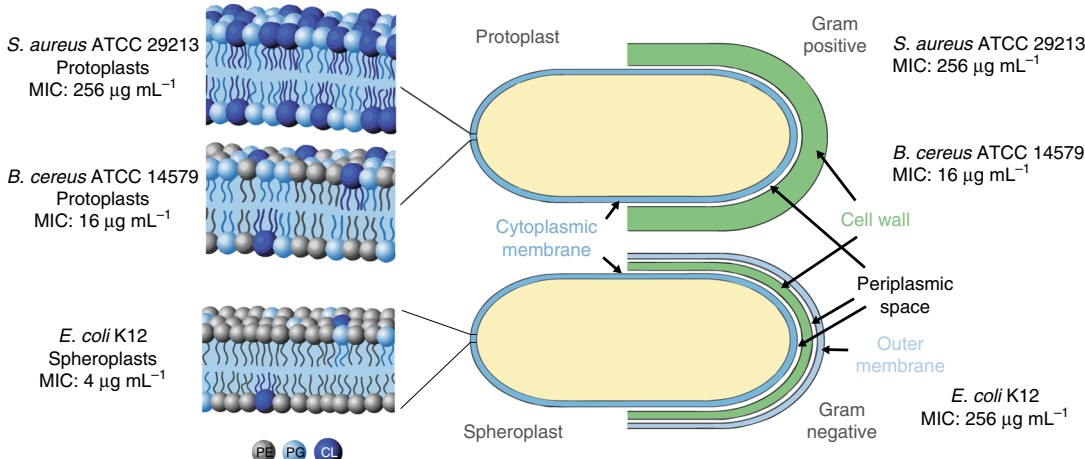

**Fig. 3** Effect of **1** in Gram-positive bacteria/protoplasts and in Gram-negative bacteria/spheroplasts. Phosphatidylethanolamine (PE)—depicted in gray; phosphatidylglycerol (PG)—depicted in light blue; cardiolipin (CL)—depicted in dark blue

of the anionic lipids (e.g., cardiolipin, CL). In the presence of cationic bactericides, *E. faecalis* behaved similarly to Gram-positive bacteria with high PE content, but the mechanism of action was unknown[18].

Compound effect on spore germination is also evaluated. When spore coat is cracked and bacterial cell membrane becomes accessible, bactericidal effect occurs (Supplementary Table 7, Supplementary Fig 10) and the spore becomes unavailable for future germination. So far, results point to a mechanism of action driven by interactions with bacterial cytoplasmic membrane.

**In silico studies on membrane destabilization**. Given the key role of cell membrane on bactericidal action, in silico studies on membrane destabilization by molecular dynamics (MD) simulations are carried out with a 1,2-dimyristoyl-*sn-glycero*-3-phosphocholine (DMPC) bilayer. Even though PC lipids are only common in eukaryotes and virtually absent in *Bacillus* membranes, they are the most reliable model system for biomolecular simulations of lipid bilayers[20]. In the context of surface-active drugs, a widespread mode of action is based on transmembrane pore formation[21]. Pore morphological complexity and size, and formation kinetics make it challenging to characterize such systems by conventional MD simulations[22–25]. In addition, it has been reported that the surface tension (ST) has an impact on the size and stability of preformed transmembrane pores[26,27]. Herein, an approach based on the effect of compound **1** on the stability of pre-formed membrane pores under different ST values is devised. Systems featuring a large pore size and different number of glycoside molecules (0%, ∼ 20 mol%, or ∼ 50 mol% of compound **1**) in the two-component bilayer are obtained following an equilibration procedure. The stability of the pre-formed pores is evaluated by monitoring the size of the transmembrane water cavity throughout the simulations without external ST (Fig. 4 and Supplementary Fig 11), and pore closure is observed after a few tens of nanoseconds. Interestingly, by increasing the molar ratio of compound **1**, pore closure occurs in shorter timescales, suggesting that its neutral head group, being less polar than DMPC zwitterion, is not able to stabilize high-energy water molecules in the pore region, in contrast to other membrane-active drugs[21]. This observation holds when an increasing ST is applied in our systems (Supplementary Fig 12) and, as expected, at higher ST values the pore closure kinetics are slower. Only at unphysical high ST conditions, which cannot be related to the bacterial cell level, the significant pore enlargement and membrane disruption are observed.

Deoxy glycoside impact on membrane structural properties is investigated by simulations of a series of glycoside-DMPC binary mixtures in a hydrated bilayer environment (glycosides **1**, **6**, and **7**), as well as octyl β-D-glucopyranoside (OG) and dodecyl β-D-glucopyranoside (DG), both exhibiting detergent properties[28,29], which effect on model PC membrane systems has been experimentally investigated[30] (Fig. 5, Supplementary Figs 13-18). To evaluate their effect on phospholipid acyl chain ordering, we compute the deuterium order parameters $|S_{CD}|$ (Fig. 5, Supplementary Figs 14 and 15). Glycosides **1**, **6**, and **7** promote an ordering of phospholipid acyl chains, this effect being slightly more pronounced for glycosides featuring less hydrophilic headgroups (Fig. 5a), whereas binary mixtures with OG or DG show a different behavior, e.g., addition of DG to the bilayer has only a negligible effect on DMPC acyl chains whereas increasing OG molar fractions promotes a decrease in DMPC acyl chain ordering (Fig. 5b), in reasonable agreement with the literature[30]. Incorporation of increasing concentrations of deoxy glycosides also promotes a thickening of the membrane (Supplementary Fig 16), with the glycoside headgroups populating outer regions of the bilayer (Supplementary Figs 17 and 18). Therefore, the results clearly show a different behavior for the deoxy glycosides when compared with alkyl glucoside detergents and highlight that sugar deoxygenation modulates the effect of these molecules over a lipid bilayer mimicking human cell membrane, thus giving molecular insights into the low toxicity observed in Caco-2 cells.

**Interaction of alkyl deoxy glycosides with PE bilayers**. In the quest for deoxy glycoside mode of action, the crucial findings rely on the selectivity for *Bacillus* spp., cytoplasmic membrane composition, and the absence of specific protein targets, converging into a phospholipid molecular target, as cell membranes with exposed PE showed the highest susceptibility. A unique biophysical feature of PE is its propensity to adopt an inverted hexagonal ($H_{II}$) phase while PC lipids form the most stable bilayer (lamellar) phases, namely the fluid $L_\alpha$ phase. Progressive increase of stress in membrane curvature resulting from $L_\alpha$ to $H_{II}$ phase transition can induce structural defects in membrane morphology, increased permeability, and, eventually, loss of integrity. In vivo, these processes can lead to cell death[31]. To understand the key role of PE, glycosides **1** and **13**, both with similar surface activity[6] but with contrasting antimicrobial activity over *B. cereus* (Table 1), are investigated. Interaction with PE bilayers is first detected by monitoring turbidity changes at 450 nm induced by these glycosides in PE large unilamellar vesicles (LUVs). Both compounds

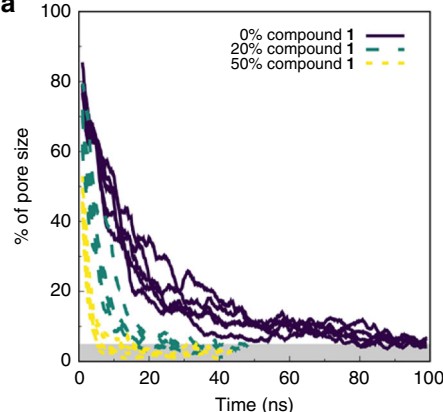

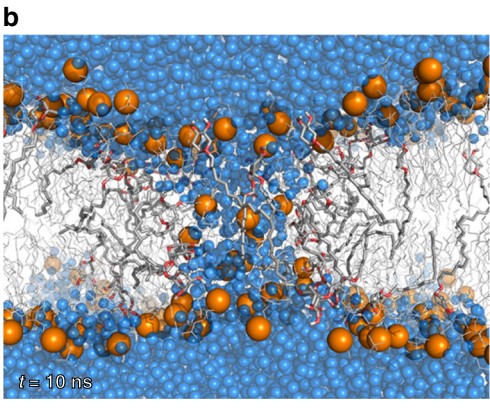

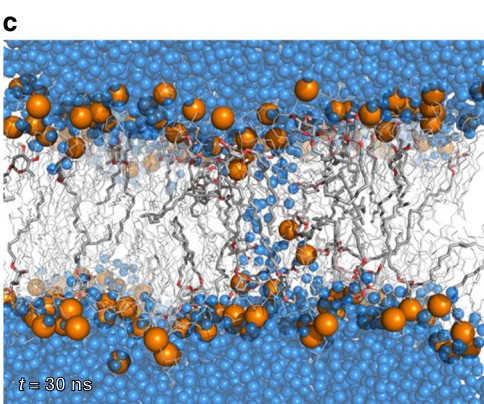

**Fig. 4** Effect of compound **1** on the stability of pre-formed membrane pores. **a** Estimated size of pre-formed membrane pores over time without external surface tension. The five replicates of each system, namely pure DMPC (control) and mixtures containing ∼20% or 50% of compound **1** are presented. A full pore corresponds to a region with ∼1800 water molecules between leaflets. The gray region represents very small pore sizes and, in many cases, complete closure of the pore. Pore size was estimated using the number of water molecules in the interior of the transmembrane cavity. The data are shown as a floating window average (2 ns) for clarity. A graphical representation of **b** a large pore and **c** a small pore is also shown for a 20% molar ratio mixture. DMPC molecules are shown as thin lines while those of compound **1** are shown as thicker sticks. Water molecules and phosphorus atoms are shown as blue and orange spheres, respectively

induce a concentration-dependent turbidity increase, reflecting that significant membrane interaction with the compounds can be easily detected in the range of concentrations where the most promising agents are active. The two compounds studied show completely different trends, pointing to different interaction

modes (Fig. 6a). PE lamellar to hexagonal phase transition is then examined by a well-established method, the temperature dependence of the steady-state fluorescence anisotropy of an appropriate probe incorporated in PE liposomes[32,33]. This method is chosen among other major techniques to study lipid polymorphism, because it allows to carry out the experiment in conditions close to physiological[34], using compound:lipid ratios and absolute compound concentration that can be directly related to those that correspond to the biological studies described above in this study. In order to select an appropriate probe, controls are performed with multilamellar vesicles (MLVs) composed exclusively of 1-palmitoyl-2-oleoyl-*sn-glycero*-3-phosphoethanolamine (POPE). Three probes already reported in the literature as detectors of phase transition are tested: 1,2-dipalmitoyl-*sn-glycero*-3-phosphoethanolamine-N-(7-nitro-2-1,3-benzoxadiazol-4-yl) (NBD-DPPE)[35], 1,2-dimyristoyl-*sn-glycero*-3-phosphoethanolamine-N-(lissamine rhodamine B sulfonyl) (Rh-PE)[35,36], and 1-(4-trimethylammoniumphenyl)-6-phenyl-1,3,5-hexatriene *p*-toluenesulfonate (TMA-DPH)[33]. Rh-PE and TMA-DPH detect the lamellar to hexagonal phase transition at 71 °C, in agreement with hexagonal phase transition temperature ($T_H$) values reported in the literature[35–37]. In both cases, $T_H$ is observed as a minimum on the thermotropic dependence of fluorescence anisotropy curve where, in the first place, anisotropy decreases reflecting the increase of the motional freedom of the probe in the lamellar phase, abruptly increasing after the transition to $H_{II}$[35]. TMA-DPH is selected for subsequent studies, as it yields better defined curves and more precise $T_H$ values (Supplementary Figs 19 and 20). Moreover, the fluorescence anisotropy of TMA-DPH also reports on the PE order at the level of the ester/first acyl chain methyl groups, giving precious information on the effects of the compounds on the organization of the bilayer in the *L*α phase and thus a deeper understanding on the biophysical mechanism behind their action.

By using MLVs composed of POPE/Soy:PE 3:1 mol:mol, the fluorescence anisotropy of the TMA-DPH probe over a temperature range of 24–44 °C demonstrates a parabolic profile with a well-defined minimum at 35.3 ± 0.6 °C (Fig. 6), corresponding to the lamellar to $T_H$. Identical MLVs suspensions containing compound **1** or **13** (50 μM) are also tested and compared (Fig. 6). For temperatures preceding $T_H$, the presence of compound **1** leads to a decrease in fluorescence anisotropy values when compared with those observed for both control and compound **13**, reflecting a decrease in the order of the lipid matrix, as opposed to compound effect on bilayers of the human cell abundant PC, a phospholipid without tendency to form hexagonal phases. On the other hand, compound **13** seems to induce order in the vesicles, when compared to the control. Moreover, although this compound does not affect $T_H$ of the PE binary mixture ($T_H = 35.2 ± 1.1$ °C), compound **1** clearly decreases $T_H$ ($T_H = 32.2 ± 0.8$ °C), increasing the propensity of PE mixture to reorganize into a hexagonal phase. This observation also corroborates the different trends of turbidity and distinct mechanisms underlying membrane interactions with each compound. These assays complement each other as turbidity changes reflect the behavior of liposomes upon increasing glycoside concentration, whereas in the fluorescence anisotropy test the glycoside is previously incorporated in the liposomes, at the highest concentration tested.

A membrane passive permeability assay is chosen to directly assess both the ability of compounds **1** and **13** to interact with POPE vs. POPC bilayers, and to evaluate their membrane permeabilizing activity toward these two different glycerophospholipids. LUVs suspensions with encapsulated carboxyfluorescein at a high concentration (40 mM) will have very low fluorescence intensity due to self-quenching. As it crosses the

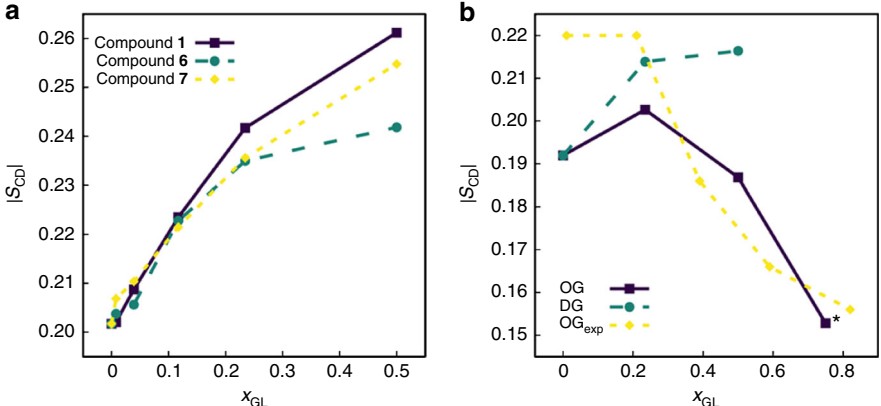

**Fig. 5** Effect of alkyl glycosides on phospholipid acyl chain ordering. **a** Deuterium order parameters at the sixth methylene group of DMPC, averaged over both sn-1 and sn-2 chains, as a function of the respective glycoside molar fraction for compounds **1**, **6**, and **7**. **b** Control deuterium order parameters at the seventh methylene group of DMPC in the presence of octyl β-D-glucopyranoside (OG) or dodecyl β-D-glucopyranoside (DG). Experimental data shown have been determined for multilamellar bilayer dispersions containing DPPC and OG at 45 °C[30]. *For $x_{OG} = 0.75$, the bilayer structure distorts significantly by the end of the simulation

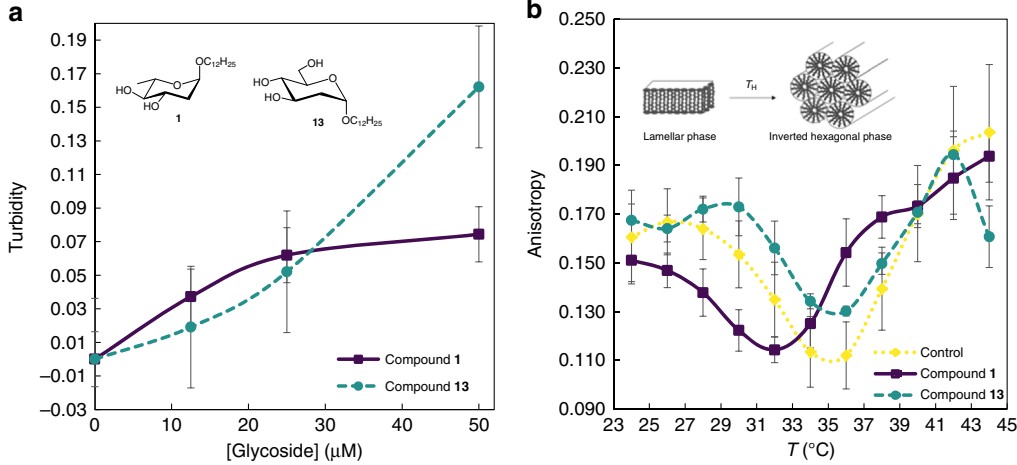

**Fig. 6** Glycoside-membrane interactions and induction of hexagonal phases. **a** Turbidity increase upon addition of compounds **1** and **13** to LUVs of POPE at 35 °C. Results are given as a quotient between the absorbance of LUVs in the presence of glycoside minus its intrinsic absorbance ($A-B$) and the absorbance of LUVs in the presence of ethanol ($A_O$), thus expressing exclusively the changes in turbidity of LUVs caused by their interaction with glycosides. Results are expressed as mean ± SE ($n = 3$). **b** Steady-state fluorescence anisotropy of TMA-DPH in binary mixtures of POPE/Soy:PE 3:1 in the presence or absence of compounds **1** and **13** (50 µM). Each dot represents the mean of three replicates for control and compound **1** and five replicates for compound **13**, expressed as mean ± SE

lipid membrane, carboxyfluorescein will be at very low concentration in the outer buffer and, consequently, will be relieved from fluorescence self-quenching. As a result, the kinetics of leakage can be monitored as an increase of fluorescence intensity over time. These membrane permeability curves are shown in Fig. 7a. From the results obtained for the membrane passive permeability it is clear that compound **1** is the most active one, and that POPE membranes are extremely susceptible to this compound, which induces a complete release (~ 100 % of leakage) of encapsulated carboxyfluorescein, whereas compound **13** only seems to slightly perturb the membrane. In this case, only ~ 10 % leakage is obtained (see also Fig. 7b). Such results show that compounds **1** and **13** interact with the POPE membrane differently. Although compound **13** seems to promote a minor perturbation of membrane organization, compound **1** leads to a more drastic reorganization of the lipid membrane with the concomitant full release of carboxyfluorescein, in very good agreement with the previous results. Moreover, POPC LUVs are more resilient than

POPE liposomes to the action of compound **1**, as a total membrane leakage of ~ 30 % is observed for POPC membranes during the course of the experiment in opposition to the ~ 100 % of leakage for POPE liposomes. Even the $L_{max}$ value obtained from the fit (Fig. 7b) is < 50% for POPC. These results strongly suggest that compound **1** interacts differently with PC or PE membranes. The low permeabilizing activity in PC bilayers seems to be in good agreement with the results of the MD simulations. The formation of pores owing to the localized transition from lamellar to an inverted hexagonal-like phase in POPE membranes (or at least of high-curvature stress areas) as a result of the interaction with compound **1** would have as outcome the complete release of carboxyfluorescein. On the other hand, all other situations where an incomplete release of carboxyfluorescein is obtained, surely, do not involve a lamellar-to-hexagonal phase transition or a high-curvature stress, but most probably only a smaller perturbation on the packing of the lipids within the bilayer. The presence of 1.2 % (v/v) of ethanol only leads to a

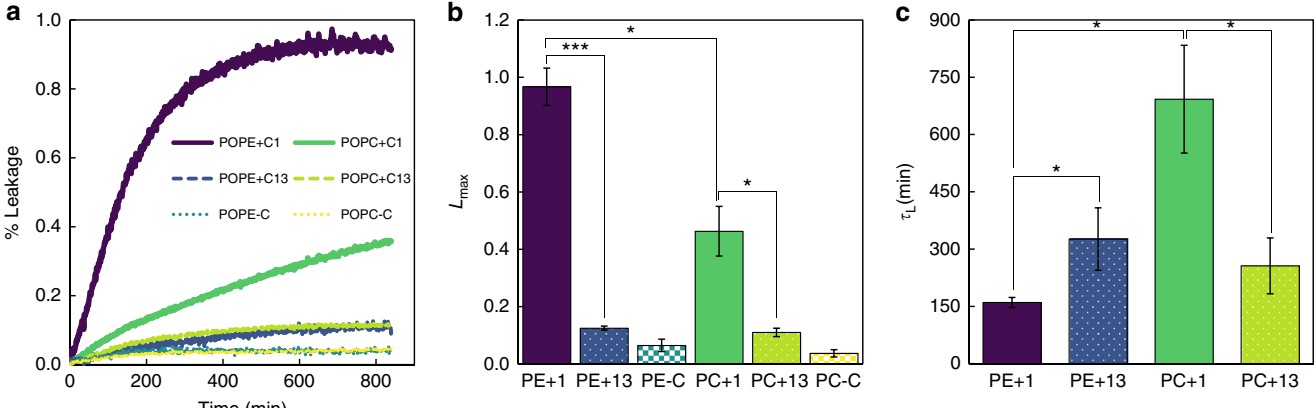

**Fig. 7** Membrane permeabilizing activity of **1** and **13** towards POPE vs. POPC membranes. **a** Representative membrane permeability curves of POPE and POPC LUVs in the absence or presence of compound **1** and **13** at 50 µM. The leakage of carboxyfluorescein from LUVs over time, which is a measure of membrane permeability due to the action of **1** and **13**, was obtained through the variation of carboxyfluorescein fluorescence intensity at excitation and emission wavelengths of 495 and 535 nm, respectively, according to Equation 1 (see Methods section). The nonlinear fitting of an exponential law (Equation 2, Methods section) allows to quantitatively describe the maximum extent and kinetics of the process. In **b** and **c** the curve fitting parameters, maximum membrane leakage ($L_{max}$) and **c** leakage time ($\tau_L$) are shown as average ± SD of three independent experiments (Student's $t$-test: $^*p < 0.05$; $^{***}p < 0.001$)

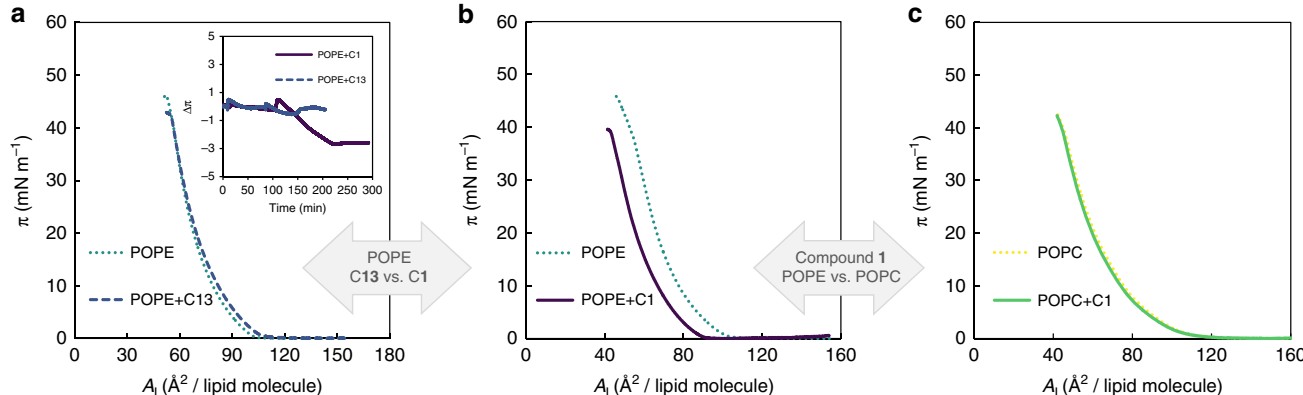

**Fig. 8** Effect of **1** and **13** on the compression isotherm of POPE or POPC. **a** Compression isotherms of a POPE monolayer before (light blue line) and after (dark blue line) incubation with compound **13** (0.2 µM). **b** Compression isotherms of a POPE monolayer before (light blue line) and after (purple line) incubation with compound **1** (0.2 µM). **c** Compression isotherms of a POPC monolayer before (yellow line) and after (green line) incubation with compound **1** (0.2 µM). Changes in surface pressure ($\Delta\pi$) of preformed POPE monolayers at $\pi \sim 25$ mN m$^{-1}$ induced by compound **1** (purple line) or **13** (dark blue) are also shown (inset in **a**). The compounds were present at the concentration 0.1 µM after the first addition (5 min) and at concentration 0.2 µM after the second addition (90 min). The maximum values of $\Delta\pi$ are 2.61 ± 0.25 mN m$^{-1}$ for compound **1** and 0.15 ± 0.06 mN m$^{-1}$ for compound **13** ($n = 3$)

negligible leakage (1–2 %) during the course of the experiment (14 h).

From the analysis of the permeability curves the time of leakage, $\tau_L$, is also obtained (Fig. 7c). The fastest process is the permeabilization of POPE by **1** ($\tau_L$ of about 160 min), whereas the slowest activity is also for compound **1**, but when added to POPC ($\tau_L$ of about 692 min). Thus, the interaction of **1** is much faster with PE bilayers than PC, suggesting a higher affinity for the PE bilayer. On another hand, compound **13** has intermediate values of $\tau_L$ that are not markedly different for both lipid bilayers, of about 326 min for POPE and 256 min for POPC. Thus, the speed of interaction of this compound (and thus possibly its affinity) is similar for both lipids. Both compounds have a hydrophobic dodecyl chain, so it is expected that they present some interaction with both glycerophospholipid bilayers. Overall, these results strongly support that **1** but not **13** has higher affinity for PE than PC, and also that the mode of action of **1** behind its antimicrobial activity toward microorganisms with high levels of exposed PE

indeed involves its specific interaction with PE leading to membrane permeabilization.

To further support the distinctive interaction of compound **1** with PE, surface pressure ($\pi$) measurements are carried out using a Langmuir trough. The effect of compounds **1** and **13** on the compression isotherm of POPE molecules at the air/water interface is assessed (Fig. 8a, b). Compression isotherms of POPE alone show a liquid expanded–liquid condensed transition around 37 mN m$^{-1}$ and collapse upon reaching surface pressures of ~ 54 mN m$^{-1}$, which is in good agreement with other reports from literature[38,39]. However, the compression curves recorded after injection of compound **1** (Fig. 8b) show a clear shift to the left, i.e., lower mean molecular area per lipid ($A$/lipid) values. To attain the same $A$/lipid, the difference (drop in this case) in $\pi$ can be as high as 15 mN m$^{-1}$. Such shift may be a consequence of altered packing properties of the POPE monolayer and/or a lesser number of POPE molecules available for the formation of the monolayer. The local action of compound **1** may trigger the

increase of monolayer curvature, possibly with the formation of invaginations. These per se can justify the shift of the curves toward lower $\pi$-values. If a fraction of the POPE molecules in these high-curvature areas undergoes a lamellar-to-hexagonal phase transition, they no longer reside at the air–water interface plane, as they will tend to aggregate and possibly precipitate. In opposition, compression isotherms of POPE recorded after the incubation with compound **13** (Fig. 8a) exhibit a slight shift to the right, i.e., higher $A$/lipid molecule values. This observation is consistent with the insertion/incorporation of compound **13** into the lipid membrane without triggering any remarkable membrane reorganization. Moreover, preformed POPE monolayers at ~ 25 mN m$^{-1}$ are incubated either with compounds **1** or **13** and the variation in $\pi$ is monitored over time until equilibration, i.e., the interaction/incorporation of the compounds on the PE mono-layer is followed over time (inset in Fig. 8a). Notably, the addition of compound **1** at a concentration as small as 0.2 µM leads to a marked decreased in $\pi$ of almost 3 mN m$^{-1}$, especially when compared with the effect of compound **13** at the same concentration, which is unable to promote any obvious change in the final $\pi$-value. Such a large effect for this concentration range denotes a specific/high-affinity interaction between the antimicrobial agent with the lipid molecules rather than a general interaction of an amphiphilic compound, such as a fatty acid or a fatty acid derivative, with the lipid monolayer[40–42]. Changes induced by compound **13** are < 0.5 mN m$^{-1}$ and after equilibration $\pi$ returns to its initial value.

Compression isotherms of pure POPC monolayers, contrary to POPE monolayer, reveal that POPC remains in the liquid expanded phase during all compression cycle, and that it collapses around 44 mN m$^{-1}$, which is in fine agreement with literature[39,43]. Compression isotherms of POPC monolayers (Fig. 8c) before and after injection of compound **1** also support a unique interaction between the compound and POPE. As mentioned, upon the incubation of POPE monolayers with compound **1** the compression curves are clearly shifted to lower $A$/lipid. However, in what concerns POPC, the curves obtained after incubation with compound **1** are practically superimposed with those acquired in the absence of this compound. Although cell membranes are more complex than liposomes, these results point to structural tendencies of lipids while interacting with glycoside **1** and support the distinct behavior of compounds **1** and **13** against *B. cereus*. The bioactivity relates to PE reorganization thereby promoting lamellar to hexagonal phase transition, which emerges as the mechanism underlying mem-brane disruption and bactericidal activity of compound **1** over *B. cereus*. The lack of activity of **13** over *B. cereus* is consistent with the biophysical studies herein presented. Worth mentioning, PE constitutes only 23% of mammalian plasma membrane phos-pholipids, and is mostly confined to the inner monolayer[44,45], therefore not directly accessible to the antimicrobial agent, highlighting the importance of this mode of action with therapeutic potential. In general, it is assumed that membrane interactions involving charge neutralization have an important role in antimicrobial peptides (AMPs) mode of action, as many of these peptides are cationic or present a highly cationic surface that promotes binding to bacterial membrane anionic phospho-lipids, such as lipopolysaccharides, phosphatidylglycerol, or CL[45–47]. However, cinnamycin, a peptide that targets PE through a very strong binding, while presenting very weak binding to PC membranes, has net formal charge zero at physiological pH[46,48,49]. In agreement, our active compound, bearing no charge, also targets PE. A literature survey on the antimicrobial susceptibility of *B. cereus* (ATCC 14579) (reviewed in Supple-mentary Table 8) shows that most bactericides reported belong to the traditional classes of antibiotics, such as penicillins,

amphenicols, fluoroquinolones, and rifamycins[50–56]. However, these classes now face substantial resistance problems to a number of bacteria and research on alternative solutions is not ubiquitous. Although *B. cereus* (ATCC 14579) has been reported to be susceptible to membrane-targeting agents, such as polymyxin B[54] and daptomycin[55], which act by membrane permeabilization and depolarization, their MICs are disappoint-ingly high and present high host toxicity. In addition, these AMPs have considerable limitations, such as low in vivo stability due to degradation by proteases, extensive serum binding, and loss of antimicrobial activity in the presence of physiological concentra-tion of salts[57], characteristic of peptide drugs. The antimicrobial lead series herein discussed and the mode of action relying on PE binding represents a significant step toward antimicrobials with new mechanisms of action.

## Discussion

One aspect that makes targeting cell membrane so appealing is its essentiality, which does not depend on the metabolic state of the cell. Unfortunately, chemotypes with bacterial membrane dis-ruption properties are often disregarded due to concerns over selectivity. In our view, and others[58], membrane-targeting anti-biotics have been underexploited. The well-known differences between eukaryotic and prokaryotic cell membrane composition encouraged us to search for leads selectively interacting with prokaryotic cell membranes, succeeding with the development of the present family of dodecyl deoxy glycosides. The glycone conducting to the highest bactericide activity over *B. anthracis* strains and *B. cereus* with low toxicity and selectivity over pro-karyotics is that of dodecyl 4,6-dideoxy-α-D-*xylo*-hexopyranoside (**11**) but deoxy *C*-glycosides of D-series with α-configuration should also be highlighted regarding activity and toxicity. By having a determinant role in PE-rich membrane interaction, bacteria resistance to these bactericides is avoided, as cell envel-ope ultrastructures cannot easily change without substantial loss of function. In summary, we set forth a creative approach relying on a multidisciplinary strategy and covering chemistry, biology, and biophysics, which led to the discovery of sugar-based anti-microbials acting over PE-rich membrane microbes by targeting membrane lipid polymorphism.

## Methods

**Chemical synthesis**. Reactions were conducted under nitrogen using dried solvents, unless otherwise stated, and monitored by thin-layer chromatography carried out on silica-coated aluminum foil plates (visualized by UV or 10% H$_2$SO$_4$ in MeOH). Compounds were purified by column chromatography and their structural characterization was accessed by NMR and high-resolution mass spectrometry analyses. Specific NMR assignements were determined by multidimensional and decoupling experiments. $^{1}$H and $^{13}$C NMR spectra for final compounds are provided in Supplementary Figs 21–39. Detailed procedures and full characterization of all compounds are available in Supplementary Methods.

**Antimicrobial assays and bacterial strains**. For results presented in Table 1, the broth microdilution method on Müller–Hinton medium was employed to deter-mine the MICs, according to the CLSI guidelines[15]. Three strains of *B. anthracis* (pathogenic strain—a human isolate, ovine strain—an animal isolate, and Sterne strain, all three from the biobank of the National Health Institute Doutor Ricardo Jorge (INSA), confirmed as *B. anthracis* strains by real-time PCR and biochemical tests), *B. cereus* (ATCC 10876), and *E. faecalis* (ATCC 29212) were used. Com-pounds were incubated with bacteria for 16 h at 37 °C before determining the MIC, which is defined as the lowest concentration of the tested compounds at which no bacterial growth was observed. For all biological studies described below, the strains used were *B. cereus* ATCC 14579 and/or *S. aureus* ATCC 25923, unless otherwise stated. For determination of bactericidal activity, MICs were determined by a modified microdilution method also according to CLSI[15]. The MBC was deter-mined in Müller–Hinton Agar (MHA) and bacterial susceptibility was assessed at 17 h. MBC was considered the lowest concentration of compound resulting in bacterial death (average of at least three independent experiments). A very similar protocol was used for the time-kill assay, where the incubation was performed with

the desired starting bacterial population ($10^6$, $10^7$, and $10^8$ cells mL$^{-1}$). Then, at desired incubation times, aliquots were retrieved, serial diluted, and transferred to MHA, where the number of colony-forming units (CFU) was determined (experimental details for MIC, MBC, and time-killing curves are available in Supplementary Methods).

**Resistance development.** Resistant mutants to the major families of antibiotics were created using a variant of the gradient-plate methodology[59], for the following antibiotics of diverse classes: erythromycin (macrolide), penicillin, vancomycin (glycopeptide), ciprofloxacin (quinolone), and tetracycline (Sigma-Aldrich). The plates were inoculated with 100 μL of 0.5 McFarland bacterial suspension of *B. cereus* ATCC 14579 strain, carefully dispersed on the agar surface using spreading spheres. The plates were incubated at 30 °C during 24 h. Afterwards, the colony closest to the higher concentration in the gradient was collected, resuspended and re-inoculated in new fresh gradient plates. This procedure was repeated for 15 days.

**Cytotoxicity.** Caco-2 cell culture and viability assays were performed using a pre-established methodology[60]. Caco-2 cells (ATCC® HTB-37™), from human color-ectal adenocarcinoma and tested negative for mycoplasma (MycoAlert™ mycoplasma detection kit from Lonza Walkersville, Inc.), were grown in Dulbecco's modified Eagle's medium high glucose supplemented with 10% fetal bovine serum, 2 mM glutamine, 100 U mL$^{-1}$ penicillin, and 100 μg mL$^{-1}$ streptomycin, in a humidified incubator at 37 °C with a 5% $CO_2$ atmosphere. The cells were trypsinized twice a week with trypsin/EDTA (0.05%/0.02%) and the medium was also changed twice a week. Compound antiproliferative effect was performed using the MTT assay by using 96-well microplates, where 20,000 cells per well were seeded. After 24 h, compounds were added to the plate in concentrations ranging from 0.1 to 100 μM. In order to evaluate an eventual solvent cytotoxicity, a control with the used solvent was carried out in all experiments. After the appropriate incubation time, freshly prepared MTT (0.5 mg mL$^{-1}$) was added to each well, and the plate was incubated for 3–4 h in a humidified atmosphere with 5% $CO_2$, at 37 °C. Then, the MTT supernatant solution was removed and the blue formazan crystals were dissolved by adding EtOH/dimethyl sulfoxide (1:1) (100 μL). The optical density was measured at 570 nm and 650 nm on a microplate reader Victor3 (PerkinElmer Life Sciences).

**Preparation of protoplasts and spheroplasts.** Bacterial spheroplasts and protoplasts [lipopolysaccharide (LPS) and peptidoglycan-free bacteria] were prepared by collecting bacterial cells by centrifugation ($10,000 \times g$ for 10 min at 4 °C) and washing with 10 mM Tris-HCl (pH 8) at 4 °C. Cells were resuspended and supplemented with sucrose (0.6 M) at room temperature. Lysozyme (0.5 mg mL$^{-1}$) (Sigma-Aldrich) was added, very slowly aiming to decrease the aggregation of plasts, after 30 min of incubation. The plasts (used at the same concentration as in the biological activity assay) were incubated with the desired compound and with polymyxin B as a positive control, at various concentrations.

**Morphologic evaluation by AFM.** Samples containing *B. cereus* ATCC 14579 ($10^6$ CFU mL$^{-1}$) in MH medium were incubated with compound **1** at various concentrations (0, 8, and 16 μg mL$^{-1}$) and after 2 h aliquots were taken. A drop (40 μL) containing the bacteria was deposited onto freshly cleaved mica surfaces for 30 min, gently washed with Milli-Q water, and dried under mild nitrogen flux. The surface was examined ex situ, at ambient temperature (21 °C), using Nanoscope IIIa multimode AFM (Digital Instruments, Veeco, Santa Barbara, CA) and etched silicon tips (TESP-V2, Bruker) with a resonance frequency of ca. 300 kHz. Images were acquired with scan rates between 1.2 Hz and 1.5 Hz. It is worth to note that washing and drying steps, as well as tip repetitive scanning did not influence the results presented in this work.

**Differential metabolomics analysis.** Bacterial metabolic response to **1** was evaluated using Phenotype MicroArrays (Biolog), assessing the metabolism of 95 different carbon sources by *B. cereus*. PM1 microplates were prepared with PM inoculating fluid (100 μL), sealed, shielded from light, and incubated at 35 °C with orbital shaking. Compound **1** (at the final concentration of 16, 8, 4, and 0 μg mL$^{-1}$) was added at 1 h of incubation. Bacterial growth was monitored by OD$_{590nm}$ at time 0 min, at 30 min, at 60 min after incubation, then hourly for 27 h.

**Susceptibility of Bacillus spores.** Production of spores from *B. cereus* ATCC 14579 was carried out by incubation at 37 °C in sporulation medium [CCY—*Bacillus* t Sporulation media:[61] Acid casein hydrolysate (1 g L$^{-1}$), 1 g L$^{-1}$ bacto casitone (pancreatic casein hydrolysate), 0.4 g L$^{-1}$ yeast extract, 20 mg L$^{-1}$ L-glutamine, 0.06% (v/v) glycerol, 1.77 g L$^{-1}$ KH$_2$PO$_4$, 4.53 g L$^{-1}$ K$_2$HPO$_4$, 0.5 mM MgCl$_2$.6H$_2$O, 0.01 mM MnCl$_2$.4H$_2$O, 0.05 mM ZnCl$_2$, 0.2 mM CaCl$_2$.6H$_2$O, 0.05 mM FeCl$_3$.6H$_2$0] for 96 h. Purification of the spores was achieved by centrifugation at 4 °C, $15,000 \times g$ and washing 10 times with Milli RO water. For preparation to the susceptibility assays, the spores were suspended in 1 mM PBS and 0.01% Tween 20, then submitted to heat treatment at 70 °C for 30 min. Germination was achieved in AGFK buffer (30 mM L-asparagine, 5,6 mM D-Glucose, 5,6 mM D-

Fructose, 20 mM KCl, and 50 mM Tris-HCl, pH 8.4) at 37 °C. Susceptibility to compound **1** was assessed at all the preparation steps, 20 min and 2 h after the germination process.

**Computational methods.** DFT calculations were performed with Gaussian 09[62]. All geometry optimizations and subsequent frequency calculations were carried out with PBE0[63]/6-311 G**[64]. Solvent effects were accounted by the integral equation formalism of the polarizable continuum model[65] on the electronic density [steered molecular dynamics (SMD)][66]. Molecular mechanics/molecular dynamics (MM/MD) simulations were performed using GROMACS version 4.0.7[67]. The GROMOS 54A7 force field[68] was employed, along with the GROMOS 56A$_{CARBO}$ force field[69] to model carbohydrates. All systems were solvated with adequate number of SPC[70] water molecules in periodic tetragonal boxes. The simulations were performed in a constant number, pressure and temperature (NPT) ensemble with temperature and pressure being kept at 308 K and 1 bar, respectively. The ST simulations were performed by applying different negative lateral pressure values in the *xy* plane of the simulation box. Initial configurations for the membrane pore and glycoside/phospholipid bilayer simulations were pre-equilibrated, energy minimized and initialized following stepwise procedures as described in Supplementary Methods, together with full computational details.

**Steady-state fluorescence anisotropy.** POPE and 1,2-diacyl-*sn-glycero*-3-phosphoethanolamine from soybean (SoyPE) were purchased from Avanti Polar Lipids (Alabaster, AL, USA) and Sigma-Aldrich (St. Louis, MO, USA), respectively. TMA-DPH was purchased from Molecular Probes, Inc. (Eugene, OR, USA). HEPES buffer contains 10 mM HEPES, 150 mM NaCl, 1 mM EDTA (pH 7.4). LUVs were extruded with Whatman® nucleopore polycarbonate filters (Sigma-Aldrich, St. Louis, USA) in a mini-extruder (Avanti Polar Lipids, Alabaster, USA). Turbidity was measured in a spectrophotometer (Beckam Instruments, Inc., Fullerton, USA), whereas steady-state fluorescence anisotropy measurements were made using a Horiba Jobin Yvon Spex Fluorolog 3-22/Tau 3 spectrofluorometer (Kyoto, Japan). MLVs containing 0.4 mol% TMA-DPH were obtained by co-dissolving the lipids (POPE:soyPE 3:1), probe, and glycoside in chloroform, followed by solvent evaporation under nitrogen stream and high vacuum desiccation. Dried lipid films were then hydrated with 10 mM HEPES buffer (pH 7.4) to the desired final concentration (0.5 mM lipid, and 0/50 μM glycoside). Lipid suspensions were subjected to five cycles of freezing and thawing at 25 °C. Anisotropy was measured by exciting samples at 360 nm and collecting emission at 430 nm, using 2.7 nm slit width. Temperature was maintained with an isothermal bath and was monitored with a thermistor, inserted into the sample chamber.

**Turbidity measurements.** MLVs composed exclusively of 2 mM POPE were prepared as above and then converted into LUVs by 17 passages through two stacked 0.1 μm polycarbonate filters in a mini-extruder, at 30 °C. LUVs were titrated with the compounds **1** or **13** in ethanol, or ethanol (control) and, after 5 min, turbidity was measured with a double beam spectrophotometer, at 35 °C, as the ratio between the absorbance of LUVs suspension at 450 nm in the presence of glycoside minus its intrinsic absorbance ($A$–$B$), and the absorbance of LUVs in the presence of the same volume of ethanol ($A_0$). Zero absorbance was set with LUVs, prior to addition of glycoside.

**Membrane leakage assay.** The effect of both compounds **1** and **13** on lipid bilayers passive permeability was assessed by monitoring the leakage of intraliposomal carboxyfluorescein, by the increase of its fluorescence intensity over time. LUVs suspensions composed of POPE or POPC were prepared in Hepes 10 mM, pH 7.4, containing carboxyfluorescein at a concentration of 40 mM. Non-encapsulated carboxyfluorescein was separated from the LUVs by ultracentrifugation at $150,000 \times g$ for 2 h. After discarding the supernatant, the pellet containing the vesicles was resuspended in Hepes 50 mM (isosmotic with the previous buffer), pH 7.4, and centrifuged again at $150,000 \times g$ for 2 h. The supernatant is rejected and the pellet resuspended in the desired volume of Hepes 50 mM, pH 7.4. The LUVs were then distributed in a 96-well clear bottom microplate to a final lipid concentration of ~ 0.5 mM and a final volume of 250 μL. The release of carboxyfluorescein was measured as the variation of fluorescence intensity at excitation and emission wavelengths of 495 and 535 nm, respectively, with a cutoff filter at 530 nm, using a microplate reader (Gemini EM Microplate Reader, Molecular Devices), at 30 ± 1 °C. The fluorescence intensity readings were performed for ca. 15 min with the LUVs suspension before adding the compounds to confirm liposome stability. Compounds **1** and **13** were added from an ethanol stock solution and the same volume of ethanol [1.2 % (v/v)] was added to the control wells. Finally, Triton X-100 was added to a final concentration of 0.5% (v/v) to obtain the fluorescence intensity corresponding to the complete release of carboxyfluorescein. Quantification of leakage was

determined according to Equation 1:

$$\%\text{Leakage} = \frac{F_t - F_0}{F_{100} - F_0} \qquad (1)$$

where $F_t$ is the fluorescence value over time after blank subtraction, $F_0$ corresponds to the initial fluorescence of the vesicle suspension, and $F_{100}$ stands for the fluorescence value after the addition of Triton X-100. Experimental leakage curves were fitted with an exponential equation (Equation 2):

$$L = L_{\max}[1 - \exp(-t/\tau_L)] \qquad (2)$$

where $L_{\max}$ stands for the maximum leakage associated to each kinetic constant, $t$ is the time after the addition of the compound and $\tau_L$ is leakage time.

**Monolayer measurements**. The interaction of compounds **1** and **13** with phospholipids was also evaluated using lipid monolayers formed at the air–water interface. A computer-controlled Langmuir trough (Kibron, µTrough G1, Helsinki, Finland with a resolution of 0.06 mN m$^{-1}$) was used to record compression isotherms ($\pi$–A) and ST variation over time of pure POPE and POPC monolayers in the absence and in the presence of the compounds. All glassware and Langmuir trough components used were rinsed thoroughly with ethanol and purified water (Millipore) before each experiment.

Aliquots of lipids in chloroform were spread, drop-by-drop, onto the subphase (Hepes 10 mM, pH 7.4, 150 mM NaCl). In order to guarantee the complete evaporation of the solvent, monolayers were let to settle for at least 15 min before experiment start.

Compression isotherms were obtained for POPC and POPE at 30 ± 1 °C and stirring in the absence and presence of compound **1** and for POPE also in presence of **13**. Compression isotherms of each monolayer were performed until two subsequent compression curves overlapped, after which the compounds were added to the subphase to a maximum compound: lipid ratio of 1:10. Then, compression curves were acquired until two successive curves superimposed again. Barrier speed during compression was set to ~ 5 cm$^2$ min$^{-1}$. To further test the interaction of the compounds with POPE, after an equilibrium $\pi$ of about 25 mN m$^{-1}$ was attained for a preformed POPE monolayer, the variation of surface pressure ($\pi$) was recorded over time upon two successive injections of 0.1 µM of compounds **1** or **13** into the subphase, allowing for equilibration between additions.

For further details, see Supplementary Methods.

## Data availability

The authors declare that all data supporting the findings of this study are available within the paper and its Supplementary Information files, and from the authors upon reasonable request.

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

## Acknowledgements

The European Union is gratefully acknowledged for the support of the project "Diagnostic and Drug Discovery Initiative for Alzheimer's Disease" (D3i4AD), FP7-PEOPLE-2013-IAPP, GA 612347. We thank the Management Authorities of the European Regional Development Fund and the National Strategic Reference Framework for the support of the Incentive System—Research and Technological Development Co-Promotion FACIB Project number 21457. Fundação para a Ciência e a Tecnologia is also acknowledged for the support of projects UID/Multi/0612/2013, FCT/UID/ Multi/04046/2013, IF/00808/2013/CP1159/CT0003, PTDC/BBBBQB/6071/2014, as well as for the post-doc grant SFRH/BPD/42567/2007 (A.M.), the Ph.D. grants SFRH/BDE/51998/2012 (C.D.), and SFRH/BDE/51957/2012 (J.P.P.), both co-sponsored by CIPAN, and also for the Ph.D. grant SFRH/BD/116614/2016 (R.N.).

## Author contributions

A.P.R. designed the project and most of the experiments, and coordinated the work carried out by all contributors, supervising also the synthetic work. C.D., R.N., M.T.B.-S., A.F.A., N.M.X., and P.S. were involved in the synthesis of alkyl deoxy glycosides and their *C*-glycosyl analogs. The role of A.M. and A.P. is related to the antimicrobial activity assays over *B. anthracis*, whereas toxicity was assessed by J.P. and M.C., supervised by N.C. Biology experiments were performed by J.P. supervised by R.D. and R.T. J.P., M.S.S., and A.V. were involved in bacteria morphological assessment and the AFM images were obtained by J.P. and A.V. Computational studies were carried out by R.N. and D.V.-V., supervised by M.M. R.F.M.A. has conceived the biophysical approach and experiments realized by C.D. and J.T.M. M.C.O. carried out the HRMS experiments for compounds' characterization. A.P.R. and C.D., J.P., R.N., and R.A. took an active part in writing the paper.

## Additional information

**Competing interests:** The authors declare no competing interests.

