## [Peer Review File · Nature Communications]

Reviewers' comments:

Reviewer #1 (Remarks to the Author):

The manuscript by Rauter et al describes their intensive investigation of the differential mechanism of action of a suite of 6-deoxy lipoglycosides against *B. anthracis* and related model strains. The authors 1) synthesize a suite of lipoglycoside analogs and test their antibiotic activity and cytotoxicity, 2) examine differential activity against cell wall deficient Gram-positive and Gram-negative strains, and 3) test hypothetical mechanism of action by in silico modeling and assays in unilamellar and multilamellar vesicles. Their conclusion, based on the presented data is that the 6-deoxy series of compounds selectively targets membrane lipid polymorphism in a phosphatidylethanolamine (PE) dependent manner. Overall, this is some rather nice work and the experiments that are presented have been carried out with a reasonable degree of rigor. I believe that the conclusions, if properly substantiated, would be of significant interest to the journal broad readership, in particular, to those with interests in antibiotic activity and microbial cell biology. Still, I have several scientific concerns that keep me from fully buying the authors conclusions and these would need to be addressed before publication in the journal:

First, the authors provide only a slight SAR for their lipoglycosides. They present just one glycoside containing a 6-hydroxyl group as the potential inactive variant. It would be helpful to have a couple such 6-hydroxy containing compounds so as to ascertain the strength of this key pharmacophore position. The current synthesis should be readily amenable to making a couple additional control compounds. This is especially necessary because the SAR on the rest of the molecule is flat - there is negligible difference between the 6-deoxy compounds (difference is within assay error).

Second, the mechanism that the authors put forth based on their anisotropy experiments seems to suggest PE binding, but this is insufficiently explored. The authors mention AMPs that bind PE, such as cinnamycin and duramycin. PE binding by these class I antibiotics has been validated in micellar systems via ITC (see Machaidze et al, *Biochemistry*, 2003, 42 (43), pp 12570–12576 and citing). Could the authors provide analogous experiments to tightly validate a PE-dependent binding mechanism? Additionally, the cinnamycin binding experiments contradict the authors statements about PE-binding AMPs (lines 296-8) being largely electrostatic (cinnamycin has a net formal charge of 0 at physiological pH). Also, cinnamycin activity results in resistant mutants that down regulate membrane associated PE. How do the authors justify a PE-based mechanism with no analogous mutations? I wonder if the authors might be able to test activity against mutants deficient in PE based on their mechanism. Overall, a secondary experiment to validate the PE mechanism would be heavily preferred.

Last, additional references would help in a number of places:

Line 149 - needs a reference for Phenotype MicroArray Approach

Line 203 - needs references regarding activity of other compounds.

Line 296 - regarding AMPs and mechanism of action.

Also, Fig. 3, which summarizes some essential data is incomplete. It would be good to include all activities discussed in this section in the figure (e.g. activity against *Staph* protoplasts and *E. faecalis* - Table S6).

Reviewer #2 (Remarks to the Author):

In this manuscript, authors developed new chemicals for bactericides. Atomic force microscopy and molecular dynamics simulation, and fluorescence anisotropy were used to study molecular mechanism. The manuscript is well written. My main concern is about the importance of this research. To support the importance, authors can include a side-by-side comparison for their new compounds and other bactericides.

Other issues:

1. In Fig. 4, the authors show that increasing the molar ratio of compound 1, transmembrane pore closure occurs in shorter time-scales, indicating that this compound does not stabilize pre-formed transmembrane pores in PC bilayers. However, as we know, the membrane tension is critical on the evolution of the transmembrane pores, e.g., to close, stabilize, or dilate [Biophys. J., (2004) 86, 2156; Front. Mol. Neurosci. (2016) 9, 136]. It seems that in the simulations, the membrane tension is so small that the transmembrane pore tends to close. It would be better to consider different membrane tension conditions.

2. The results in Fig. 4 show that the compound 1 is helpful to close the transmembrane pore in contrast to other membrane-active drugs. What is the difference between the compound 1 and other membrane-active drugs regarding the stability of membrane? What would be the molecular mechanism for healing the transmembrane pore?

3. In Figs 4B & C, authors should use a different color for compound 1 in contrast to the PC molecules.

Editor and referee's letter replying point by point to the reviewer comments

Reviewer 1

The manuscript by Rauter et al describes their intensive investigation of the differential mechanism of action of a suite of 6-deoxy lipoglycosides against *B. anthracis* and related model strains. The authors 1) synthesize a suite of lipoglycoside analogs and test their antibiotic activity and cytotoxicity, 2) examine differential activity against cell wall deficient Gram-positive and Gram-negative strains, and 3) test hypothetical mechanism of action by in silico modeling and assays in unilamellar and multilamellar vesicles. Their conclusion, based on the presented data is that the 6-deoxy series of compounds selectively targets membrane lipid polymorphism in a phosphatidylethanolamine (PE) dependent manner. Overall, this is some rather nice work and the experiments that are presented have been carried out with a reasonable degree of rigor. I believe that the conclusions, if properly substantiated, would be of significant interest to the journal broad readership, in particular, to those with interests in antibiotic activity and microbial cell biology. Still, I have several scientific concerns that keep me from fully buying the authors conclusions and these would need to be addressed before publication in the journal

1. First, the authors provide only a slight SAR for their lipoglycosides. They present just one glycoside containing a 6-hydroxyl group as the potential inactive variant. It would be helpful to have a couple such 6-hydroxy containing compounds so as to ascertain the strength of this key pharmacophore position. The current synthesis should be readily amenable to making a couple additional control compounds. This is especially necessary because the SAR on the rest of the molecule is flat - there is negligible difference between the 6-deoxy compounds (difference is within assay error).

Reply: According to this request, we have added a new set of compounds (including their synthesis and characterization) to those previously given in the manuscript, namely the dodecyl 2-deoxy C-glycoside **14** analogue to **13** and the 4-deoxy O-glycoside **15**, both of them with the 6-OH group in their structure. We have added the 4,6-dideoxy glycoside **11** and the β -anomers of the 2,6-dideoxy and the 4,6-dideoxy glycosides, namely compounds **5** and **12**, aiming to give some new examples of active and inactive deoxy glycosides, thus reinforcing the role of the glycone structure to increase antimicrobial activity. Accordingly, Fig. 1 was changed with the introduction of the structure of such compounds, and SI contains now their synthesis and structure characterization. Maria Teresa Blazquez Sanchez was added as co-author, since she synthesized newly introduced compounds.

The antimicrobial activity of compounds **1**, **5**, **11**, **12**, **13**, **14** and **15** over *B. cereus* and *B. anthracis* strains Sterne, ovine and pathogenic was evaluated using the broth microdilution method on Müller-Hinton medium. Also cytotoxicity over Caco-2-cells was evaluated. So, with the new added structures we followed reviewer 1 requests, by adding inactive structures (e.g. compounds **5** and **12**), showing that the anomeric configuration and the deoxygenation pattern are paramount for the bioactivity, and the presence of 6-OH decreases compound bioactivity, as deduced by comparing compound pairs **14/10** and **15/11** (Fig 1), including **13/9** for *B. anthracis* Sterne and *B. cereus*. We discovered that the 4,6-dideoxy pattern is the most promising one leading to the most active compound in all bacteria and strains tested. Compound **13** revealed inactive for *B. cereus* and *B. anthracis* Sterne, as expected, but active over *B. anthracis* pathogenic and ovine. This was not detected in our preliminary assays with compound **13**, carried out by the paper disc diffusion method over *Bacillus cereus*, and also not expected because *B. cereus* genome is known to be very similar to that of *B. anthracis*, as reported in the literature and cited in the manuscript. Therefore, compound **13** was investigated, together with the lead compound **1**, to evaluate the mechanism of action. In fact the battery of biophysical tests carried out demonstrated that compound **13** has a completely different behavior than that of the lead compound, which bioactivity results from targeting membrane lipid polymorphism, namely the propensity of phosphatidylethanolamine to undergo a lamellar to hexagonal phase transition.

2. Second, the mechanism that the authors put forth based on their anisotropy experiments seems to suggest PE binding, but this is insufficiently explored. The authors mention AMPs that bind PE, such as cinnamycin and duramycin. PE binding by these class I antibiotics has been validated in micellar systems via ITC (see Machaidze et al, *Biochemistry*, 2003, 42 (43), pp 12570–12576 and citing). Could the authors provide analogous experiments to tightly validate a PE-dependent binding mechanism? Additionally, the cinnamycin binding experiments contradict the authors statements about PE-binding AMPs (lines 296-8) being largely electrostatic (cinnamycin has

a net formal charge of 0 at physiological pH). Also, cinnamycin activity results in resistant mutants that down regulate membrane associated PE. How do the authors justify a PE-based mechanism with no analogous mutations? I wonder if the authors might be able to test activity against mutants deficient in PE based on their mechanism. Overall, a secondary experiment to validate the PE mechanism would be heavily preferred.

Reply: We fully agree with this Reviewer 1 comment, and therefore we provide, in the revised version novel data to validate the PE-dependent mechanism of action. We now include two additional types of approaches: 1) membrane permeabilizing activity of compounds **1** and **13** towards PC versus PE membranes and 2) surface pressure experiments in PE and PC monolayers. In both cases the data strongly support the proposed mechanism and previous conclusions (new Figs 7 and 8). The new results now presented, in particular the membrane permeabilizing activity, show unequivocally that PE membranes are highly susceptible to the active compound **1**, but not to compound **13**. Moreover, they also show very clearly that the impact on PE membranes of the active compound is not only much stronger, but is also much faster, when compared to its impact on PC membranes, which may also help to explain the lack of resistance development and corroborates the MD simulations in PC membranes. In addition, the effects observed suggest the induction of highly curved membrane regions, which in the extreme situation would correspond to the formation of inverted hexagonal phase as observed by fluorescence anisotropy in PE membranes, but when operating in a more localized fashion can induce e.g., vesiculation. Joaquim T. Marquês was added as co-author, since he performed the new biophysical experiments.

In the revised version of the manuscript, one can now read:

“A membrane passive permeability assay was used to directly assess both the ability of compounds **1** and **13** to interact with POPE versus POPC bilayers, and to evaluate their membrane permeabilizing activity towards these two different glycerophospholipids. LUVs suspensions with encapsulated carboxyfluorescein at a high concentration (40 mM) will have very low fluorescence intensity due to self-quenching. As it crosses the lipid membrane, carboxyfluorescein will be at very low concentration in the outer buffer and, consequently, will be relieved from fluorescence self-quenching. As a result, the kinetics of leakage can be monitored as an increase of fluorescence intensity over time. These membrane permeability curves are shown in Fig. 7A.

From the results obtained for the membrane passive permeability it is clear that compound **1** is the most active one and that POPE membranes are extremely susceptible to this compound, that induces a complete release (~100 % of leakage) of encapsulated carboxyfluorescein, whereas compound **13** only seems to slightly perturb the membrane. In this case only ~10 % leakage was obtained (see also Fig. 7B). Such results show that compounds **1** and **13** interact with the POPE membrane differently. While compound **13** seems to promote a minor perturbation of membrane organization, compound **1** leads to a more drastic reorganization of the lipid membrane with the concomitant full release of carboxyfluorescein, in very good agreement with the previous results. Moreover, POPC LUVs were more resilient than POPE liposomes to the action of compound **1**, since a total membrane leakage of ~30 % was observed for POPC membranes during the course of the experiment in opposition to the ~100 % of leakage for POPE liposomes. Even the L_{max} value obtained from the fit (Fig. 7B) is less than 50% for POPC. These results strongly suggest that compound **1** interacts differently with PC or PE membranes. The low permeabilizing activity in PC bilayers seems to be in good agreement with the results of the MD simulations. The formation of pores owing to the localized transition from lamellar to an inverted hexagonal-like phase in POPE membranes (or at least of high curvature stress areas) as a result of the interaction with compound **1** would have as outcome the complete release of carboxyfluorescein. On the other hand, all other situations where an incomplete release of carboxyfluorescein was obtained, surely, do not involve a lamellar-to-hexagonal phase transition or a high curvature stress, but most probably only a smaller perturbation on the packing of the lipids within the bilayer. The presence of 1.2 % (v/v) of ethanol only leads to a negligible leakage (1-2 %) during the course of the experiment (14 h).

Fig. 7. Representative membrane permeability (carboxyfluorescein leakage) curves of POPE and POPC LUVs in the absence or presence of compound **1** and **13** at 50 μM (panel A). In panels B and C the curve parameters, maximum membrane leakage (L_{max}) (B) and leakage time (τ_L) (C), are shown as average \pm s.d. of at least three independent experiments.

From the analysis of the permeability curves the time of leakage, τ_L , was also obtained (Fig. 7C). The fastest process was the permeabilization of POPE by **1** (τ_L of about 160 min), whereas the slowest activity was also for compound **1**, but when added to POPC (τ_L of about 692 min). Thus the interaction of **1** is much faster with PE bilayers than PC, suggesting a higher affinity for the PE bilayer. On another hand, compound **13** had intermediate values of τ_L that were not markedly different for both lipid bilayers, of about 326 min for POPE and 256 min for POPC. Thus the speed of interaction of this compound (and thus possibly its affinity) is similar for both lipids. Both compounds have a hydrophobic dodecyl chain, so it is expected that they present some interaction with both glycerophospholipid bilayers. Overall these results strongly support that compound **1** but not **13** has higher affinity for PE than PC, and also that the mode of action of **1** behind its antimicrobial activity towards microorganisms with high levels of exposed PE indeed involves its specific interaction with PE leading to membrane permeabilization.

To further support the distinctive interaction of compound **1** with PE, surface pressure (π) measurements were carried out using a Langmuir trough. The effect of compounds **1** and **13** on the compression isotherm of POPE molecules at the air/water interface was assessed (Fig. 8, A and B). Compression isotherms of POPE alone showed a liquid expanded-liquid condensed transition around 37 mN/m and collapsed upon reaching surface pressures of ~ 54 mN/m, which is in good agreement with other reports from literature^{38,39}. However, the compression curves recorded after injection of compound **1** (Fig. 8A) show a clear shift to the left, i.e. lower mean molecular area per lipid (A/lipid) values. To attain the same A/lipid , the difference (drop in this case) in π can be as high as 15 mN/m. Such shift may be a consequence of altered packing properties of the POPE monolayer and/or a lesser number of POPE molecules available for the formation of the monolayer. The local action of compound **1** may trigger the increase of monolayer curvature, possibly with the formation of invaginations. These per se can justify the shift of the curves towards lower π values. If a fraction of the POPE molecules in these high curvature areas undergo a lamellar-to-hexagonal phase transition they no longer reside at the air-water interface plane, as they will tend to aggregate and possibly precipitate. In opposition, compression isotherms of POPE recorded after the incubation with compound **13** (Fig. 8B) exhibit a slight shift to the right, i.e. higher A/lipid molecule values. This observation is consistent with the insertion/incorporation of compound **13** into the lipid membrane without triggering any remarkable membrane reorganization.

Moreover, preformed POPE monolayers at ~ 25 mN/m were incubated either with compounds **1** or **13** and the variation in π was monitored over time until equilibration, i.e., the interaction/incorporation of the compounds on the PE monolayer was followed over time (inset in Fig. 8A). Notably, the addition of compound **1** at a concentration as small as 0.2 μM leads to a marked decreased in π of almost 3 mN/m, especially when compared with the effect of compound **13** at the same concentration, which was unable to promote any obvious change in the final π value. Such a large effect for this concentration range denotes a specific/high affinity interaction between the antimicrobial agent with the lipid molecules rather than a general interaction of an amphiphilic compound, such as a fatty acid or a fatty acid derivative, with the lipid monolayer⁴⁰⁻⁴². Changes induced by compound **13** were smaller than 0.5 mN/m and after equilibration π returned to its initial value.

Compression isotherms of pure POPC monolayers, contrary to POPE monolayer, revealed that POPC remains in the liquid expanded phase during all compression cycle and that it collapses around 44 mN/m, which is in fine agreement with

literature^{39,43}. Compression isotherms of POPC monolayers (Fig. 8C) before and after injection of compound **1** also support a unique interaction between the compound and POPE. As mentioned, upon the incubation of POPE monolayers with compound **1** the compression curves are clearly shifted to lower $A/lipid$. However, in what concerns POPC, the curves obtained after incubation with compound **1** are practically superimposed with the ones acquired in the absence of the compound.

Fig. 8. Compression isotherms of POPE monolayers before (black line) and after (gray line) incubation with compound **13** ($0.2 \mu\text{M}$) (panel A) and **1** ($0.2 \mu\text{M}$) (panel B) and of POPC monolayers before (black line) and after (gray line) incubation with compound **1** ($0.2 \mu\text{M}$). Changes in surface pressure ($\Delta\pi$) of preformed POPE monolayers at $\pi \sim 25 \text{ mN/m}$ induced by compound **1** (black line) or **13** (gray line) are also shown (inset in panel A). The compounds were at $\sim 0.1 \mu\text{M}$ after the first addition (1st) and at $\sim 0.2 \mu\text{M}$ after the second addition (2nd). Curves shown are representative of at least three independent experiments.

Although cell membranes are more complex than liposomes, these results point to structural tendencies of lipids while interacting with glycoside **1** and support the distinct behavior of compounds **1** and **13** against *Bacillus cereus*. The bioactivity relates to PE reorganization thereby promoting lamellar to hexagonal phase transition, which emerges as the mechanism underlying membrane disruption and bactericidal activity of compound **1** over *B. cereus*. The lack of activity of **13** over *B. cereus* is consistent with the biophysical studies herein presented. Worth mentioning, PE constitutes only 23% of mammalian plasma membrane phospholipids, and is mostly confined to the inner monolayer^{44,45}, therefore not directly accessible to the antimicrobial agent, highlighting the importance of this mode of action with therapeutic potential.

We also wish to acknowledge the Reviewer by noting that the importance of electrostatics in the membrane binding and mechanism of action was not stated clearly and contradicts the results with cinammycin. This is now corrected in the revised version. We have also included references Machaidze et al, *Biochemistry*, 2003, 42 (43), pp 12570–12576 and Machaidze et al, *Biochemistry*, 2002, 41 (6), pp 1965–1971 (refs 48 and 49), and added the following sentence to the manuscript:

“Generally, it is assumed that membrane interactions involving charge neutralization play an important role in antimicrobial peptides (AMPs) mode of action, as many of these peptides are cationic or present a highly cationic surface that promotes binding to bacterial membrane anionic phospholipids, such as lipopolysaccharides, phosphatidylglycerol or cardiolipin⁴⁵⁻⁴⁷. However, cinammycin, a peptide that targets PE through a very strong binding, while presenting very weak binding to PC membranes, has net formal charge zero at physiological pH.^{46,48,49} In agreement, our active compound, bearing no charge, also targets PE. “

As the Reviewer pointed out, PE is a nearly zwitterionic lipid at that pH. In agreement, our active compound, also targeting PE, is a neutral molecule. Nonetheless, all the biophysical results presented by us in this revised version, as stated above, highlight that the interaction of compound **1** with PE is not merely of hydrophobic nature, as the outcomes of insertion into

PE are completely different from those of insertion into PC membranes. The turbidity measurements (formerly given in SI, now inserted in the manuscript in Fig. 6) show that both the active compound **1** and inactive compound **13** insert into PE membranes. However, in case of compound **1** a plateau is reached for a concentration value close to MIC. The other biophysical assays show that the membrane is disrupted, probably involving morphological alterations, counteracting the increase in particle size owed to compound incorporation, justifying the plateau. For compound **13**, the almost linear trend up to the maximum concentration of compound tested in the turbidity experiments shows an efficient incorporation, probably due to hydrophobic/non-specific interactions, without significant impact on membrane organization. The curve for compound **13** is almost superimposed with that for compound **1** in the linear part, suggesting that a binding per se would not clarify the mechanism proposed, hence our choice of alternative methods (membrane leakage and monolayer studies) to further support the PE-based mechanism in addition to the turbidity and the fluorescence anisotropy assays.

3. Last, additional references would help in a number of places:

1. Line 149 - needs a reference for Phenotype MicroArray Approach

R: Reference 16 was added:

16. A. Shea, M. Wolcott, S. Daeﬂer, D. A. Rozak. (2012) Biolog Phenotype Microarrays. In: Microbial Systems Biology. Methods in Molecular Biology (Methods and Protocols) A. Navid Ed. (Humana Press, 2012), vol. 881, Totowa, NJ.

2. Line 203 - needs references regarding activity of other compounds.

R: Reference 21 was added

21. K. A. Brogden. Antimicrobial peptides: Pore formers or metabolic inhibitors in bacteria? *Nat. Rev. Microbiol.* 3, 238–250 (2005).

3. Line 296 - regarding AMPs and mechanism of action.

R: References 48, 49 were added (see above, please) as well as refs 46 and 47:

46. R. M. Epanand, C. Walker, R. F. Epanand, N. A. Magarvey, Molecular mechanisms of membrane targeting antibiotics. *Biochim. Biophys. Acta* 1858 (5), 980–987 (2016).

47. S. Omardien, S. Brul, S. A. Zaat. Antimicrobial Activity of Cationic Antimicrobial Peptides against Gram-Positives: Current Progress Made in Understanding the Mode of Action and the Response of Bacteria. *Front. Cell Dev. Biol.* 4, 111 (2016).

4. Also, Fig. 3, which summarizes some essential data is incomplete. It would be good to include all activities discussed in this section in the figure (e.g. activity against Staph protoplasts and *E. faecalis* - Table S6).

R: Fig. 3 was changed and includes now all activities discussed, as requested by the reviewer.

Reviewer 2

In this manuscript, authors developed new chemicals for bactericides. Atomic force microscopy and molecular dynamics simulation, and fluorescence anisotropy were used to study molecular mechanism. The manuscript is well written. My main concern is about the importance of this research. To support the importance, authors can include a side-by-side comparison for their new compounds and other bactericides.

Other issues:

1. In Fig. 4, the authors show that increasing the molar ratio of compound 1, transmembrane pore closure occurs in shorter time-scales, indicating that this compound does not stabilize pre-formed transmembrane pores in PC bilayers. However, as we know, the membrane tension is critical on the evolution of the transmembrane pores,

e.g., to close, stabilize, or dilate [Biophys. J., (2004) 86, 2156; Front. Mol. Neurosci. (2016) 9, 136]. It seems that in the simulations, the membrane tension is so small that the transmembrane pore tends to close. It would be better to consider different membrane tension conditions.

2. The results in Fig. 4 show that the compound **1** is helpful to close the transmembrane pore in contrast to other membrane-active drugs. What is the difference between the compound **1** and other membrane-active drugs regarding the stability of membrane? What would be the molecular mechanism for healing the transmembrane pore?
3. In Figs 4B & C, authors should use a different color for compound **1** in contrast to the PC molecules.

R: Aiming at comparing the neutral bactericide family reported in this paper with other bactericides, the following text and table 8 (in SI) were added:

A literature survey on the antimicrobial susceptibility of *B. cereus* (ATCC 14579) (reviewed in table S8) shows that most bactericides reported belong to the traditional classes of antibiotics, such as penicillins, amphenicols, fluoroquinolones and rifamycins⁵⁰⁻⁵⁶. However, these classes now face substantial resistance problems to a number of bacteria, and research on alternative solutions is not ubiquitous. While *B. cereus* (ATCC 14579) has been reported to be susceptible to membrane targeting agents, such as polymyxin B⁵⁴ and daptomycin⁵⁵, that act by membrane permeabilization and depolarization, their MICs are disappointingly high and present high host toxicity. In addition, these antimicrobial peptides have considerable limitations, such as low in vivo stability due to degradation by proteases, extensive serum binding and loss of antimicrobial activity in the presence of physiological concentration of salts⁵⁷, characteristic of peptide drugs. The antimicrobial lead series herein discussed and their mode of action relying on PE binding, represents a significant step towards new antimicrobials with new mechanisms of action.

Other issues:

1. In Fig. 4, the authors show that increasing the molar ratio of compound **1**, transmembrane pore closure occurs in shorter time-scales, indicating that this compound does not stabilize pre-formed transmembrane pores in PC bilayers. However, as we know, the membrane tension is critical on the evolution of the transmembrane pores, e.g., to close, stabilize, or dilate [Biophys. J., (2004) 86, 2156; Front. Mol. Neurosci. (2016) 9, 136]. It seems that in the simulations, the membrane tension is so small that the transmembrane pore tends to close. It would be better to consider different membrane tension conditions.

R: We agree with the reviewer regarding the mentioned effect of surface tension (ST) on the stability of transmembrane pores. We added both references (refs 26 and 27) and performed additional simulations with ST values ranging from 0 to ~30 dyn/cm. The effect of ST is now discussed in the manuscript referring to a new figure in SI and the two sentences given below are introduced in the manuscript:

“This observation holds when an increasing ST is applied in our systems (Fig. S8) and, as expected, at higher ST values the pore closure kinetics are slower. Only at unphysically high ST conditions, which cannot be related to the bacterial cell level, the significant pore enlargement and membrane disruption are observed.”

2. The results in Fig. 4 show that the compound **1** is helpful to close the transmembrane pore in contrast to other membrane-active drugs. What is the difference between the compound **1** and other membrane-active drugs regarding the stability of membrane? What would be the molecular mechanism for healing the transmembrane pore?

R: An important difference between this family of compounds and other membrane-active drugs is that they bare no formal charge. For instance, many antimicrobial peptides (AMPs) exert their toxicity effect taking advantage of their positive charges, associated with a strong amphiphilic character. Compound **1**, which is also amphiphilic, is not able to stabilize high energy water molecules in the pore region. Probably, the observed healing effect comes from the fact that compound **1** headgroup is less polar than DMPC phosphate and choline groups, and in contrast with AMPs charged groups. This interpretation of the healing mechanism was added to the manuscript in the following sentence:

“Interestingly, by increasing the molar ratio of compound **1**, pore closure occurs in shorter time-scales, suggesting that its neutral head group, being less polar than DMPC zwitterion, is not able to stabilize high energy water molecules in the pore region, in contrast to other membrane-active drugs.”

3. In Figs 4B & C, authors should use a different color for compound 1 in contrast to the PC molecules.

R: In our previous images, compound 1 was absent, but we accepted the reviewer suggestion and illustrated the pore sizes with two new snapshots from the 20% system. In these new figures, the glycosides can now be clearly distinguished in thicker sticks.

REVIEWERS' COMMENTS:

Reviewer #1 (Remarks to the Author):

This is very nice work and the authors have done a very thorough job of addressing my concerns. In particular, they have rounded out the SAR with several additional lipoglycosides with the activity summarized in Table 1. Revised Fig.1 gives a nice representation of the (now exhaustive) compound scope tested. I was rather pleased to see the novel activity of compound 13 as well, really contributes to quite a nice story about these compounds that might otherwise be overlooked for their combination of somewhat intricate synthetic requirements and otherwise inauspicious biophysical properties. Experiments validating the PE interaction are also very thorough and appropriate. In particular, I appreciate the use of surface pressure measurement to assess the specificity of the PE interaction; much more robust and informative with respect to the present phenomena than the ITC experiments that I had recommended.

Minor comments:

Although the synthesis is nice work, I would relegate Scheme 1 to SI, but not essential - takes focus from the compounds.

I would include a better explanation of the dT measurement in Fig.7.

Otherwise publish as is.

Reviewer #2 (Remarks to the Author):

The authors have adequately addressed all my concerns. The manuscript is substantially improved. I recommend this paper be published in the current form.

Response to the reviewer comments:

We take this opportunity to acknowledge our reviewers for their comments that improved our manuscript, and of course, we are very grateful for having accepted our work.

Sincerely

Amélia Pilar Rauter

The corresponding author

1. Reviewer #2

Reviewer #2 (Remarks to the Author):

The authors have adequately addressed all my concerns. The manuscript is substantially improved. I recommend this paper be published in the current form.

2. Reviewer #1 (Remarks to the Author):

This is very nice work and the authors have done a very thorough job of addressing my concerns. In particular, they have rounded out the SAR with several additional lipoglycosides with the activity summarized in Table 1. Revised Fig.1 gives a nice representation of the (now exhaustive) compound scope tested. I was rather pleased to see the novel activity of compound 13 as well, really contributes to quite a nice story about these compounds that might otherwise be overlooked for their combination of somewhat intricate synthetic requirements and otherwise inauspicious biophysical properties. Experiments validating the PE interaction are also very thorough and appropriate. In particular, I appreciate the use of surface pressure measurement to assess the specificity of the PE interaction; much more robust and informative with respect to the present phenomena than the ITC experiments that I had recommended.

Minor comments: I would include a better explanation of the dT measurement in Fig.7. Otherwise publish as is.

i) Although the synthesis is nice work, I would relegate Scheme 1 to SI, but not essential - takes focus from the compounds.

R: Scheme 1 was deleted from the manuscript and given, with more details, in SI (Supplementary Fig. 6).

ii) I would include a better explanation of the dT measurement in Fig.7.

R: As requested by the reviewer we present now, in Figure 7 caption, a more detailed explanation of how we measured the leakage variation with time, and how we obtained the curve parameters Lmax and TauL.

Since there were no dT measures presented in Figure 7, the Reviewer could also have referred to Turbidity measurements, because in Figure 6 we present the relative variation in turbidity. So, we have also changed the respective sub-heading from "Vesicle aggregation" to "Turbidity measurements". Moreover, we have explained in more detail the measurements in the methods section. The following changes were made:

a) "Vesicle aggregation" was changed to "Turbidity Measurements"

b) The text describing the methodology was improved and was replaced by the following one:

MLVs composed exclusively of 2 mM POPE were prepared as above, and then converted into LUVs by 17 passages through two stacked 0.1 μm polycarbonate filters in a mini-extruder, at 30 °C. LUVs were titrated with the compounds **1** or **13** in ethanol, or ethanol (control) and, after 5 min, turbidity was measured with a double beam spectrophotometer, at 35 °C, as the ratio between the absorbance of LUVs suspension at 450 nm in the presence of glycoside minus its intrinsic absorbance (A-B), and the absorbance of LUVs in the presence of the same volume of ethanol (A0). Zero absorbance was set with LUVs, prior to addition of glycoside.